# Online Structured Meta-learning

**Huaxiu Yao**[1][*] **Yingbo Zhou**[2]**, Mehrdad Mahdavi**[1]
**Zhenhui Li**[1]**, Richard Socher**[2]**, Caiming Xiong**[2]
[1]Pennsylvania State University, [2]Salesforce Research
[1]{huaxiuyao,mzm616,zul17}@psu.edu, [2]{yingbo.zhou,cxiong}@salesforce.com, richard@socher.org

## Abstract

Learning quickly is of great importance for machine intelligence deployed in online platforms. With the capability of transferring knowledge from learned tasks, meta-learning has shown its effectiveness in online scenarios by continuously updating the model with the learned prior. However, current online meta-learning algorithms are limited to learn a globally-shared meta-learner, which may lead to sub-optimal results when the tasks contain heterogeneous information that are distinct by nature and difficult to share. We overcome this limitation by proposing an online structured meta-learning (OSML) framework. Inspired by the knowledge organization of human and hierarchical feature representation, OSML explicitly disentangles the meta-learner as a meta-hierarchical graph with different knowledge blocks. When a new task is encountered, it constructs a meta-knowledge pathway by either utilizing the most relevant knowledge blocks or exploring new blocks. Through the meta-knowledge pathway, the model is able to quickly adapt to the new task. In addition, new knowledge is further incorporated into the selected blocks. Experiments on three datasets demonstrate the effectiveness and interpretability of our proposed framework in the context of both homogeneous and heterogeneous tasks.

## 1 Introduction

Meta-learning has shown its effectiveness in adapting to new tasks with transferring the prior experience learned from other related tasks [7, 34, 38]. At a high level, the meta-learning process involves two steps: meta-training and meta-testing. During the meta-training time, meta-learning aims to obtain a generalized meta-learner by learning from a large number of past tasks. The meta-learner is then applied to adapt to newly encountered tasks during the meta-testing time. Despite the early success of meta-learning on various applications (e.g., computer vision [7, 40], natural language processing [14, 36]), almost all traditional meta-learning algorithms make the assumption that tasks are sampled from the same stationary distribution. However, in human learning, a promising characteristic is the ability to continuously learn and enhance the learning capacity from different tasks. To equip agents with such capability, recently, Finn *et al.* [10] presented the online meta-learning framework by connecting meta-learning and online learning. Under this setting, the meta-learner not only benefits the learning process from the current task but also continuously updates itself with accumulated new knowledge.

Although online meta-learning has shown preliminary success in handling non-stationary task distri-bution, the globally shared meta-learner across all tasks is far from achieving satisfactory performance when tasks are sampled from complex heterogeneous distribution. For example, if the task distri-bution is heterogeneous with disjoint modes, a globally shared meta-learner is not able to cover the information from all modes. Under the stationary task distribution, a few studies attempt to

---

[*]Part of the work was done while the author interned at Salesforce Research

address this problem by modulating the globally shared meta-learner with task-specific information [27, 41, 42, 39]. However, the modulating mechanism relies on a well-trained task representation network, which makes it impractical under the online meta-learning scenario. A recent study by [15] applied Dirichlet process mixture of hierarchical Bayesian model on tasks. However, this study requires the construction of a totally new meta-learner for the dissimilar task, which limits the flexibility of knowledge sharing.

To address the above challenges, we propose a meta-learning method with a structured meta-learner, which is inspired by both knowledge organization in human brain and hierarchical representations. When human learn a new skill, relevant historical knowledge will facilitate the learning process. The historical knowledge, which is potentially hierarchically organized and related, is selectively integrated based on the relevance to the new task. After mastering a new task, the knowledge representation evolves continuously with the new knowledge. Similarly, in meta-learning, we aim to construct a well-organized meta-learner that can 1) benefit fast adaptation in the current task with task-specific structured prior; 2) accumulate and organize the newly learned experience; and 3) automatically adapt and expand for unseen structured knowledge.

We propose a novel online structured meta-learning (**OSML**) framework. Specifically, OSML disentangles the whole meta-learner as a meta-hierarchical graph with multiple structured knowledge blocks, where each block represents one type of structured knowledge (e.g., a similar background of image tasks). When a new task arrives, it automatically seeks for the most relevant knowledge blocks and constructs a meta-knowledge pathway in this meta-hierarchical graph. It can further create new blocks when the task distribution is remote. Compared with adding a full new meta-learner (e.g., [15]) in online meta-learning, the block-level design provides 1) more flexibility for knowledge exploration and exploitation; 2) reduces the model size; and 3) improves the generalization ability. After solving the current task, the selected blocks are enhanced by integrating with new information from the task. As a result, the model is capable of handling non-stationary task distribution with potentially complex heterogeneous tasks.

To sum up, our major contributions are three-fold: 1) we formulate the problem of online meta-learning under heterogeneous distribution setting and propose a novel online meta-learning framework by maintaining meta-hierarchical graph; 2) we demonstrate the effectiveness of the proposed method empirically through comprehensive experiments; 3) the constructed meta-hierarchical tree captures the structured information in online meta-learning, which enhances the model interpretability.

## 2 Notations and Problem Settings

**Few-shot Learning and Meta-learning.** In few-shot learning, a task $\mathcal{T}_i$ is comprised of a support set $\mathcal{D}_i^{supp}$ with $N_i^{supp}$ samples (i.e., $\mathcal{D}_i^{supp} = \{(\mathbf{x}_{i,1}, \mathbf{y}_{i,1}), \ldots, (\mathbf{x}_{i,N_i^{supp}}, \mathbf{y}_{i,N_i^{supp}})\}$) and a query set $\mathcal{D}_i^{query}$ with $N_i^{query}$ samples (i.e., $\mathcal{D}_i^{query} = \{(\mathbf{x}_{i,1}, \mathbf{y}_{i,1}), \ldots, (\mathbf{x}_{i,N_i^{query}}, \mathbf{y}_{i,N_i^{query}})\}$), where the support set only includes a few samples. Given a predictive model $f$ with parameter $\mathbf{w}$, the task-specific parameter $\mathbf{w}_i$ is trained by minimizing the empirical loss $\mathcal{L}(\mathbf{w}, \mathcal{D}_i^{supp})$ on support set $\mathcal{D}_i^{supp}$. Here, the loss function is typically defined as mean square loss or cross-entropy for regression and classification problems, respectively. The trained model $f_{\mathbf{w}_i}$ is further evaluated on query set $\mathcal{D}_i^{query}$. However, when the size of $\mathcal{D}_i^{supp}$ is extremely small, it cannot optimize $\mathbf{w}$ with a satisfactory performance on $\mathcal{D}_i^{query}$. To further improve the task-specific performance within limited samples, a natural solution lies in distilling more information from multiple related tasks. Building a model upon these related tasks, meta-learning are capable of enhancing the performance in few-shot learning.

By training from multiple related tasks, meta-learning generalizes an effective learning strategy to benefit the learning efficiency of new tasks. There are two major steps in meta-learning: meta-training and meta-testing. Taking model-agnostic meta-learning (MAML) [8] as an example, at meta-training time, it aims to learn a well-generalized initial model parameter $\mathbf{w}_0^*$ over $T$ available meta-training tasks $\{\mathcal{T}_i\}_{i=1}^T$. In more detail, for each task $\mathcal{T}_i$, MAML performs one or few gradient steps to infer task-specific parameter $\mathbf{w}_i$ by using support set $\mathcal{D}_i^{supp}$ (i.e., $\mathbf{w}_i = \mathbf{w}_0 - \alpha \mathcal{L}(\mathbf{w}, \mathcal{D}_i^{supp})$). Then, the query set $\mathcal{D}_i^{query}$ are used to update the initial parameter $\mathbf{w}_0$. Formally, the bi-level optimization process can be formulated as:

$$\mathbf{w}_0^* \leftarrow \arg\min_{\mathbf{w}_0} \sum_{i=1}^T \mathcal{L}(\mathbf{w}_0 - \alpha \mathcal{L}(\mathbf{w}, \mathcal{D}_i^{supp}), \mathcal{D}_i^{query}). \tag{1}$$

In practice, the inner update can perform several gradient steps. At meta-testing time, for the new task $\mathcal{T}_{new}$, the optimal task parameter $\mathbf{w}_{new}$ can be reached by finetuning $\mathbf{w}_0^*$ on support set $\mathcal{D}_{new}^{supp}$.

**Online Meta-learning.** A strong assumption in the meta-learning setting is that all task follows the same stationary distribution. In online meta-learning, instead, the agents observe new tasks and update meta-learner (e.g., model initial parameter) sequentially. It then tries to optimize the performance of the current task. Let $\mathbf{w}_{0,t}$ denote the learned model initial parameter after having task $\mathcal{T}_t$ and $\mathbf{w}_t$ represent the task-specific parameter. Following FTML (follow the meta leader) algorithm [10], the online meta-learning process can be formulated as:

$$\mathbf{w}_{0,t+1} = \arg\min_{\mathbf{w}} \sum_{i=1}^{t} \mathcal{L}_i(\mathbf{w}_i, \mathcal{D}_i^{query}) = \arg\min_{\mathbf{w}} \sum_{i=1}^{t} \mathcal{L}_i(\mathbf{w} - \alpha\nabla(\mathbf{w}, \mathcal{D}_i^{supp}), \mathcal{D}_i^{query}), \quad (2)$$

where $\{\mathcal{L}_t\}_{t=1}^{\infty}$ represent a sequence of loss functions for task $\{\mathcal{T}_t\}_{t=1}^{\infty}$. In the online meta-learning setting, both $\mathcal{D}_i^{supp}$ and $\mathcal{D}_i^{query}$ can be represented as different sample batches for task $\mathcal{T}_i$. For brevity, we denote the inner update process as $\mathcal{M}(\mathbf{w}, \mathcal{D}_i^{supp}) = \mathbf{w} - \alpha\nabla(\mathbf{w}, \mathcal{D}_i^{supp})$. After obtaining the best initial parameter $\mathbf{w}_{0,t+1}$, similar to the classical meta-learning process, the task-specific parameter $\mathbf{w}_{t+1}$ is optimized by performing several gradient updates on support set $\mathcal{D}_{t+1}^{supp}$. Based on the meta-learner update paradigm in equation (2), the goal for FTML is to minimize the regret, which is formulated as

$$\text{Regret}_T = \sum_{i=1}^{T} \mathcal{L}_i(\mathcal{M}_i(\mathbf{w}_{0,i})) - \min_{\mathbf{w}} \sum_{i=1}^{T} \mathcal{L}_i(\mathcal{M}_i(\mathbf{w})). \quad (3)$$

By achieving the sublinear regret, the agent is able to continuously optimize performance for sequential tasks with the best meta-learner.

## 3 Online Structured Meta-learning

In this section, we describe the proposed OSML algorithm that sequentially learns tasks from a non-stationary and potentially heterogeneous task distribution. Figure 1 illustrates the pipeline of task learning in OSML. Here, we treat a meta-learner as a meta-hierarchical graph, which consists of multiple knowledge blocks. Each knowledge block represents a specific type of meta-knowledge and is able to connect with blocks in the next level. To facilitate the learning of a new task, a "search-update" mechanism is proposed. For each task, a "search" operation is first performed to create meta-knowledge pathways. An "update" operation is then performed for the meta-hierarchical graph. For each task, this mechanism forms a pathway that links the most relevant neural knowledge block of each level in the meta-hierarchical structure. Simultaneously, novel knowledge blocks may also be spawned automatically for easier incorporation of unseen (heterogeneous) information. These selected knowledge blocks are capable of quick adaptation for the task at hand. Through the created meta-knowledge pathway, the initial parameters of knowledge blocks will be then iteratively updated by incorporating the new information. In the rest of this section, we will introduce the two key components: meta-knowledge pathway construction and knowledge-block update.

### 3.1 Meta-knowledge Pathway Construction

The key idea of meta-knowledge pathway construction is to automatically identify the most relevant knowledge blocks from the meta-hierarchy. When the task distribution is non-stationary, the current task may contain distinct information, which will increase the likelihood of triggering the use of novel knowledge blocks. Because of this, a meta-knowledge pathway construction mechanism should be capable of automatically exploring and utilizing knowledge blocks depending on the task distribution. We elaborate the detailed pathway construction process in the following paragraphs.

At time step $t$, we denote the meta-hierarchy with initial parameter $\mathbf{w}_{0,t}$ as $\mathcal{R}_t$, which has $L$ layers with $B_l$ knowledge blocks in each layer $l$. Let $\{\mathbf{w}_{0b_l,t}\}_{b_l=1}^{B_l}$ denote the initial parameters in knowledge blocks of layer $l$. To build the meta-knowledge pathway, we search the most relevant knowledge block for each layer $l$. An additional novel block is further introduced in the search process for new knowledge exploration. Similar to [21], we further relax the categorical search process to a differentiable manner to improve the efficiency of knowledge block searching. For each layer $l$ with the input representation $\mathbf{g}_{l-1,t}$, the relaxed forward process in meta-hierarchical graph $\mathcal{R}^t$ is

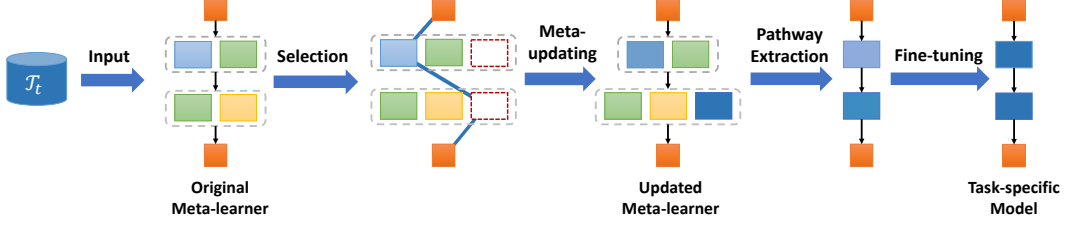

**Figure 1:** Illustration of OSML. The meta-hierarchical graph is comprised of several knowledge blocks in each layer. Different colors represent different meta-knowledge. Orange blocks denote the input and output. Given a new task $\mathcal{T}_t$, it automatically searches for the most relevant knowledge block and constructs the meta-knowledge pathway (i.e., blue line). Simultaneously, the task is encouraged to explore novel meta-knowledge blocks during the search process (i.e., the red dashed block). After building the meta-knowledge pathway, the new task is used to update its corresponding meta-knowledge blocks. The meta-updated knowledge blocks are finally used for fine-tuning and evaluation on $\mathcal{T}_t$.

formulated as:

$$\mathbf{g}_{l,t} = \sum_{b_l=1}^{B_l+1} \frac{\exp(o_{b_l})}{\sum_{b_l'=1}^{B_l+1} \exp(o_{b_l'})} \mathcal{M}_t(\mathbf{w}_{0b_l,t})(\mathbf{g}_{l-1,t}), \tag{4}$$

$$\text{where } \mathcal{M}_t(\mathbf{w}_{0b_l,t}) = \mathbf{w}_{0b_l,t} - \alpha \nabla_{\mathbf{w}_{0b_l,t}} \mathcal{L}(\mathbf{w}_{0,t}, \mathcal{D}_t^{supp}),$$

where $\mathbf{o} = \{\{o_{b_1}\}_{b_1=1}^{B_1}, \ldots, \{o_{b_L}\}_{b_L=1}^{B_L}\}$ are used to denote the importance of different knowledge blocks in layer $l$. The above equation (4) indicates that the inner update in the knowledge block searching process, where all existing knowledge blocks and the novel blocks are involved. After inner update, we obtain the task specific parameter $\mathbf{w}_t$, which is further used to meta-update the initial parameters $\mathbf{w}_{0,t}$ in the meta-hierarchical structure and the importance coefficient $\mathbf{o}$. The meta-update procedure is formulated as:

$$\mathbf{w}_{0,t} \leftarrow \mathbf{w}_{0,t} - \beta_1 \nabla_{\mathbf{w}_{0,t}} \mathcal{L}(\mathbf{w}_t, \mathbf{o}; \mathcal{D}_t^{query}),$$
$$\mathbf{o} \leftarrow \mathbf{o} - \beta_2 \nabla_{\mathbf{o}} \mathcal{L}(\mathbf{w}_t, \mathbf{o}; \mathcal{D}_t^{query}), \tag{5}$$

where $\beta_1$ and $\beta_2$ represent the learning rates at the meta-updating time. Since the coefficient $\mathbf{o}$ suggests the importance of different knowledge blocks, we finally select the task-specific knowledge block in layer $l$ as $b_l^* = \arg\max_{b_l \in [1, B_l]} o_{b_l}$. After selecting the most relevant knowledge blocks, the meta-knowledge pathway $\{\mathbf{w}_{0b_1^*,t}, \ldots, \mathbf{w}_{0b_L^*,t}\}$ is generated by connecting these blocks layer by layer.

## 3.2 Knowledge Block Meta-Updating

In this section, we discuss how to incorporate the new task information with the best path of functional regions. Following [10], we adopt a task buffer $\mathcal{B}$ to memorize the previous tasks. When a new task arrives, it is automatically added in the task buffer. After constructing the meta-knowledge pathway, the shared knowledge blocks are updated with the new information. Since different tasks may share knowledge blocks, we iteratively optimize the parameters of the knowledge blocks from low-level to high-level. In practice, to optimize the knowledge block $b_l$ in layer $l$, it is time-consuming to apply second-order meta-optimization process on $\mathbf{w}_{0b_l,t}$. Instead, we apply first-order approximation to avoid calculating the second-order gradients on the query set. Formally, the first order approximation for knowledge block $b_l^*$ in layer $l$ is formulated as:

$$\mathbf{w}_{0b_l^*,t} \leftarrow \mathbf{w}_{0b_l^*,t} - \beta_3 \sum_{k=1}^{K} \nabla_{\mathbf{w}_{b_l,k}} \mathcal{L}(\mathbf{w}_k; \mathcal{D}_k^{query}), \tag{6}$$

$$\text{where } \mathbf{w}_t = \mathbf{w}_{0,t} - \beta_4 \nabla_{\mathbf{w}} \mathcal{L}(\mathbf{w}; \mathcal{D}_k^{supp}).$$

By applying the first-order approximation, we are able to save update time while maintaining comparable performance. After updating the selected knowledge blocks, we fine-tune the enhanced meta-knowledge pathway $\{\mathbf{w}_{0b_1^*,t}, \ldots, \mathbf{w}_{0b_L^*,t}\}$ on the new task $\mathcal{T}_t$ by using both support and query sets as follows:

$$\mathbf{w}_{b_1^*,t} = \mathbf{w}_{0b_l^*,t} - \beta_5 \nabla_{\mathbf{w}} \mathcal{L}(\mathbf{w}; \mathcal{D}_t^{supp} \oplus \mathcal{D}_t^{query}). \tag{7}$$

---
**Algorithm 1** Online Meta-learning Pipeline of OSML
---
**Require:** $\beta_1$, $\beta_2$, $\beta_3$,$\beta_4$, $\beta_5$: learning rates
 1: Initialize $\Theta$ and the task buffer as empty, $\mathcal{B} \leftarrow []$
 2: **for** each task $\mathcal{T}_t$ in task sequence **do**
 3:     Add $\mathcal{B} \leftarrow \mathcal{B} + [\mathcal{T}_t]$
 4:     Sample $\mathcal{D}_t^{supp}$, $\mathcal{D}_t^{query}$ from $\mathcal{T}_t$
 5:     Use $\mathcal{D}_t^{supp}$ and $\mathcal{D}_t^{query}$ to search the functional regions $\{\mathbf{w}_{0b_1^*,t} \ldots \mathbf{w}_{0b_L^*,t}\}$ by (4) and (5)
 6:     **for** $n_m = 1 \ldots N_{meta}$ steps **do**
 7:       **for** $l = 1 \ldots L$ **do**
 8:           Sample task $\mathcal{T}_k$ that also use $\mathbf{w}_{0b_l^*,t}$ from buffer $\mathcal{B}$
 9:           Sample minibatches $\mathcal{D}_k^{supp}$ and $\mathcal{D}_k^{query}$ from $\mathcal{T}_k$
10:           Use $\mathcal{D}_k^{supp}$ and $\mathcal{D}_k^{query}$ to update the knowledge block $\mathbf{w}_{0b_l^*,t}$ by (6)
11:       **end for**
12:     **end for**
13:     Concatenate $\mathcal{D}_t^{supp}$ and $\mathcal{D}_t^{query}$ as $\mathcal{D}_t^{all} = \mathcal{D}_t^{supp} \oplus \mathcal{D}_t^{query}$
14:     Use $\mathcal{D}_t^{all}$ to finetune $\{\mathbf{w}_{0b_1^*,t} \ldots \mathbf{w}_{0b_L^*,t}\}$
15:     Evaluate the performance on $\mathcal{D}_t^{test}$
16: **end for**
---

The generalization performance of task $\mathcal{T}_t$ are further evaluated in a held-out dataset $\mathcal{D}_t^{test}$. The whole procedure are outlined in Algorithm 1.

## 4 Experiments

In this section, we conduct experiments on both homogeneous and heterogeneous datasets to show the effectiveness of the proposed OSML. The goal is to answer the following questions:

- How does OSML perform (accuracy and efficiency) compared with other baselines in both homogeneous and heterogeneous datasets?
- Can the knowledge blocks explicitly capture the (dis-)similarity between tasks?
- What are causing the better performance of OSML: knowledge organization or model capacity?

The following algorithms are adopted as baselines, including (1) Non-transfer (NT), which only uses support set of task $\mathcal{T}_t$ to train the base learner; (2) Fine-tune (FT), which continuously fine-tunes the base model without task-specific adaptation. Here, only one knowledge block each layer is involved and fine-tuned for each task and no meta-knowledge pathway is getting constructed; (3) FTML [10] that incorporates MAML into online learning framework, where the meta-learner is shared across tasks; (4) DPM [15], which uses the Dirichlet process mixture to model task changing; (5) HSML [41], which customizing model initializations by involving hierarchical clustering structure. However, the continual adaptation setting in original HSML is evaluated under the stationary scenario. Thus, to make comparison we evaluate HSML under our setting by introducing task-awared parameter customization and hierarchical clustering structure.

### 4.1 Homogeneous Task Distribution

**Dataset Description.** We first investigate the performance of OSML when the tasks are sampled from a single task distribution. Here, we follow [10] and create a Rainbow MNIST dataset, which contains a sequence of tasks generated from the original MNIST dataset. Specifically, we change the color (7 colors), scale (2 scales), and angle (4 angles) of the original MNIST dataset. Each combination of image transformation is considered as one task and thus a total of 56 tasks are generated in the Rainbow MNIST dataset. Each task contains 900 training samples and 100 testing samples. We adopt the classical four-block convolutional network as the base model. Additional information about experiment settings are provided in Appendix A.1.

**Results and Analysis.** The results of Rainbow MNIST shown in Figure 2. It can be observed that our proposed OSML consistently outperforms other baselines, including FTML, which shares the meta-learner across all tasks. Additionally, after updating the last task, we observe that the number of

knowledge blocks for layer 1-4 in the meta-hierarchical graph is 1, 1, 2, 2, respectively. This indicates that most tasks share the same knowledge blocks, in particular, at the lower levels. This shows that our method does learn to exploit shared information when the tasks are more homogeneous and thus share more knowledge structure. This also suggests that our superior performance is not because of increased model size, but rather better utilization of the shared structure.

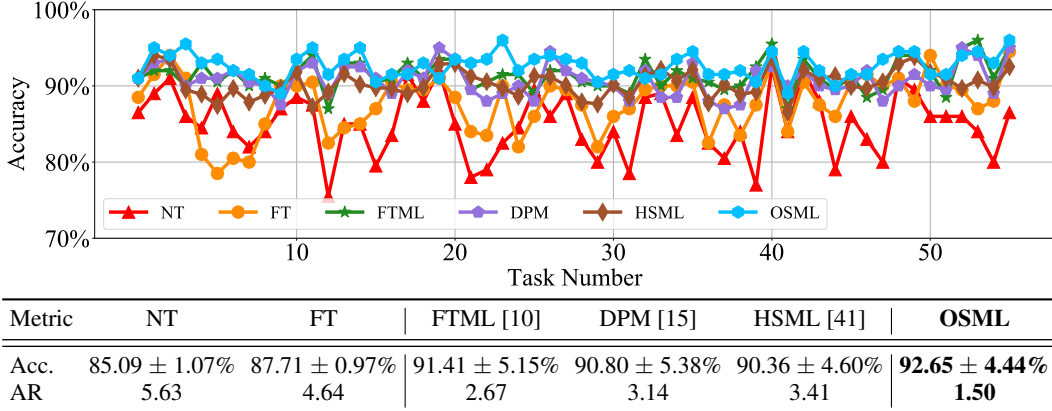

| Metric | NT | FT | FTML [10] | DPM [15] | HSML [41] | **OSML** |
|--------|-----|-----|-----------|----------|-----------|----------|
| Acc. | $85.09 \pm 1.07\%$ | $87.71 \pm 0.97\%$ | $91.41 \pm 5.15\%$ | $90.80 \pm 5.38\%$ | $90.36 \pm 4.60\%$ | $\mathbf{92.65 \pm 4.44\%}$ |
| AR | 5.63 | 4.64 | 2.67 | 3.14 | 3.41 | **1.50** |

Figure 2: Rainbow MNIST results. Top: Accuracy over all tasks; Bottom: Performance statistics. Here, Average Ranking (AR) is calculated by first rank all methods for each dataset, from higher to lower. Each method receive a score corresponds to its rank, e.g. rank one receives one point. The scores for each method are then averaged to form the reported AR. Lower AR is better.

## 4.2 Heterogeneous Task Distribution

**Datasets Descriptions.** To further verify the effectiveness of OSML when the tasks contain potentially heterogeneous information, we created two datasets. The first dataset is generated from mini-Imagenet. Here, a set of artistic filters – "blur", "night" and "pencil" filter are used to process the original dataset [15]. As a result, three filtered mini-Imagenet sub-datasets are obtained, namely, blur-, night- and pencil-mini-Imagenet. We name the constructed dataset as multi-filtered mini-Imagenet. We create the second dataset called Meta-dataset by following [37, 41]. This dataset includes three fine-grained sub-datasets: Flower, Fungi, and Aircraft. Detailed descriptions of heterogeneous datasets construction are discussed in Appendix A.2. For each sub-dataset in multi-filtered miniImagenet or Meta-dataset, it contains 100 classes. We then randomly split 100 classes to 20 non-overlapped 5-way tasks. Thus, both datasets include 60 tasks in total and we shuffle all tasks for online meta-learning. Similar to Rainbow MNIST, the four-block convolutional layers are used as the base model for each task. Note that, for Meta-dataset with several more challenging fine-grained datasets, the initial parameter values of both baselines and OSML are set from a model pre-trained from the original mini-Imagenet. We report hyperparameters and model structures in Appendix A.2.

**Results.** For multi-filtered miniImagenet and Meta-dataset, we report the performance in Figure 3 and Figure 4, respectively. We show the performance over all tasks in the top figure and summarize the performance in the bottom table. First, all online meta-learning methods (i.e., FTML, DPM, HSML, OSML) achieves better performance than the non-meta-learning ones (i.e., NT, FT), which further demonstrate the effectiveness of task-specific adaptation. Note that, NT outperforms FT in Meta-dataset. The reason is that the pre-trained network from mini-Imagnent is loaded in Meta-dataset as initial model, continuously updating the structure (i.e., FT) is likely to stuck in a specific local optimum (e.g., FT achieves satisfactory results in Aircraft while fails in other sub-datasets). In addition, we also observe that task-specific online meta-learning methods (i.e., OSML, DPM, and HSML) achieve better performance than FTML. This is further supported by the summarized performance of Meta-dataset (see the bottom table of Figure 4), where FTML achieved relatively better performance in Aircraft compared with Fungi and Flower. This suggests that the shared meta-learner is possibly attracted into a specific mode/region and is not able to make use of the information from all tasks. Besides, OSML outperforms DPM and HSML in both datasets, indicating that the meta-hierarchical structure not only effectively capture heterogeneous task-specific information, but also encourage more flexible knowledge sharing.

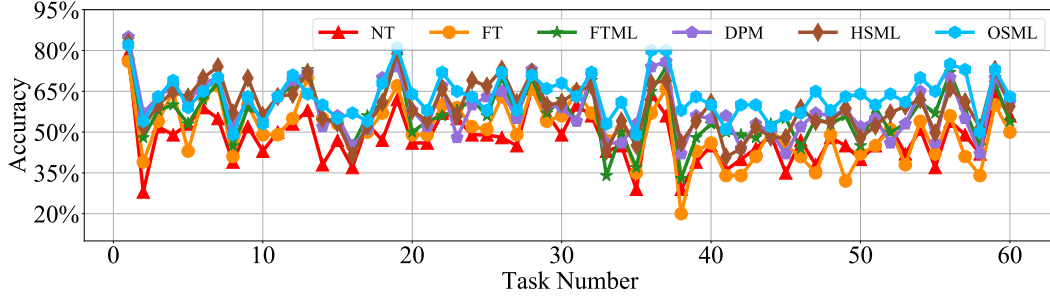

| Models | Blur Acc. | Night Acc. | Pencil Acc. | Overall Acc. | AR |
|---|---|---|---|---|---|
| NT | $49.80 \pm 3.91\%$ | $47.70 \pm 2.91\%$ | $47.55 \pm 5.18\%$ | $48.35 \pm 2.39\%$ | 5.48 |
| FT | $51.50 \pm 4.90\%$ | $49.00 \pm 3.82\%$ | $50.90 \pm 5.30\%$ | $50.47 \pm 2.73\%$ | 4.87 |
| FTML [10] | $58.90 \pm 3.52\%$ | $56.40 \pm 3.53\%$ | $54.60 \pm 5.46\%$ | $56.63 \pm 2.50\%$ | 3.50 |
| DPM [15] | $62.35 \pm 2.95\%$ | $56.80 \pm 4.28\%$ | $56.20 \pm 4.70\%$ | $58.45 \pm 2.44\%$ | 2.85 |
| HSML [41] | $62.42 \pm 3.80\%$ | $57.57 \pm 2.78\%$ | $57.88 \pm 5.00\%$ | $59.25 \pm 2.36\%$ | 2.63 |
| **OSML** | $\mathbf{64.10 \pm 3.12\%}$ | $\mathbf{65.25 \pm 3.24\%}$ | $\mathbf{60.35 \pm 3.65\%}$ | $\mathbf{63.23 \pm 2.00\%}$ | **1.67** |

Figure 3: Multi-filtered miniImagenet results. Top : classification accuracy of all tasks. Bottom : the statistics of accuracy with 95% confidence interval and average ranking (AR) for each sub-dataset

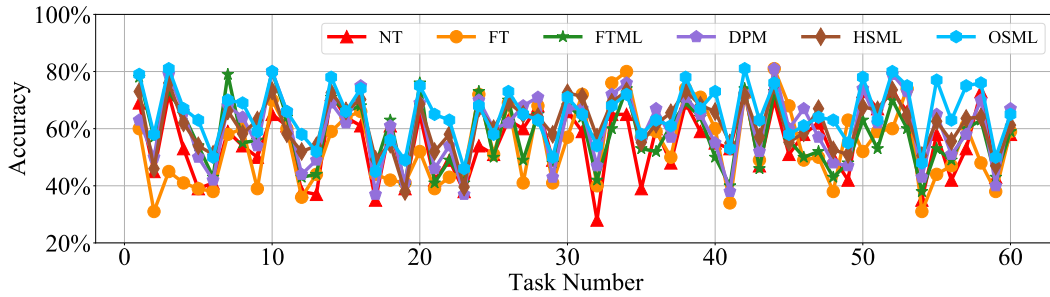

| Models | Aircraft Acc. | Flower Acc. | Fungi Acc. | Overall Acc. | AR |
|---|---|---|---|---|---|
| NT | $59.40 \pm 3.97\%$ | $60.15 \pm 5.23\%$ | $46.15 \pm 3.57\%$ | $55.23 \pm 2.98\%$ | 4.75 |
| FT | $66.45 \pm 3.80\%$ | $52.20 \pm 4.55\%$ | $42.90 \pm 3.45\%$ | $53.85 \pm 3.35\%$ | 4.47 |
| FTML [10] | $65.85 \pm 3.85\%$ | $59.05 \pm 4.83\%$ | $48.00 \pm 2.78\%$ | $57.63 \pm 2.93\%$ | 3.92 |
| DPM [15] | $66.67 \pm 4.07\%$ | $63.40 \pm 4.89\%$ | $50.15 \pm 3.53\%$ | $60.07 \pm 3.02\%$ | 3.15 |
| HSML [41] | $65.86 \pm 3.13\%$ | $63.12 \pm 3.88\%$ | $55.65 \pm 3.11\%$ | $61.54 \pm 2.22\%$ | 2.85 |
| **OSML** | $\mathbf{67.99 \pm 3.52\%}$ | $\mathbf{68.55 \pm 4.59\%}$ | $\mathbf{58.45 \pm 2.89\%}$ | $\mathbf{65.00 \pm 2.46\%}$ | **1.87** |

Figure 4: Meta-dataset Results. Top: performace of online meta-learning tasks. Bottom: performance statistics on each sub-dataset.

**Analysis of Constructed Meta-pathway and Knowledge Blocks.** In Figure 5, we analyze the selected knowledge blocks of all tasks after the online meta-learning process. Here, for each knowledge block, we compute the selected ratio of every sub-dataset in Meta-dataset (see Appendix B for results and analysis in Multi-filtered mini-Imagenet). For example, if the knowledge block 1 in layer 1 is selected 3 times by tasks from Aircraft and 2 times from Fungi. The corresponding ratios of Aircraft and Fungi are 60% and 40%, respectively. From these figures, we see that some knowledge blocks are dominated by different sub-datasets whereas others have shared across different tasks. This indicates that OSML is capable of automatically detecting distinct tasks and their feature representations, which also demonstrate the interpretability of OSML. We also observe that tasks from fungi and flower are more likely to share blocks (e.g., block 7 in layer 1). The potential reason is that tasks from fungi and flower sharing similar background and shapes, and therefore have higher probability of sharing similar representations.

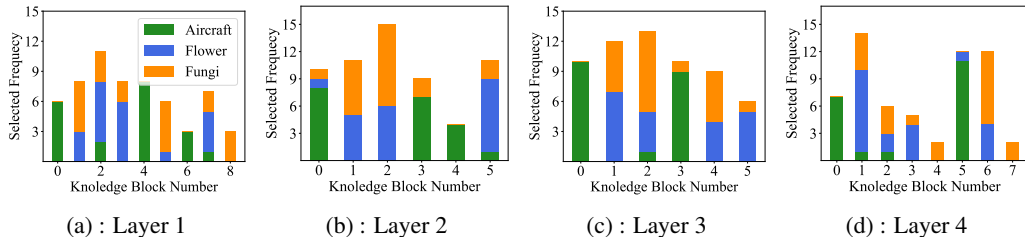

| (a) : Layer 1 | (b) : Layer 2 | (c) : Layer 3 | (d) : Layer 4 |

Figure 5: Selected ratio of knowledge blocks for each sub-dataset in Meta-dataset. Figure (a)-(d) illustrate the knowledge blocks in layer 1-4.

**Effect of Model Capacity.** Though the model capacity of OSML is the same as all baselines during task training time (i.e., a network with four convolutional blocks), OSML maintains a larger network to select the most relevant meta-knowledge pathway. To further investigate the reason for improvements, we increase the numbers blocks in NT, FT, and FTML to the number of blocks in the meta-hierarchical graph after passing all tasks. We did not include DPM and HSML since they already have more parameters than OSML. These baselines with larger representation capacity are named as NT-Large, FT-Large, and FTML-Large. We compare and report the summary of performance in Table 6 (see the results of Meta-dataset in Appendix C). First, we observe that increasing the number of blocks in NT worsens the performance, suggesting the overfitting issue. In FT and FTML, increasing the model capacity does not achieve significant improvements, indicating that the improvements do not stem from larger model capacity. Thus, OSML is capable of detecting heterogeneous knowledge by automatically selecting the most relevant knowledge blocks.

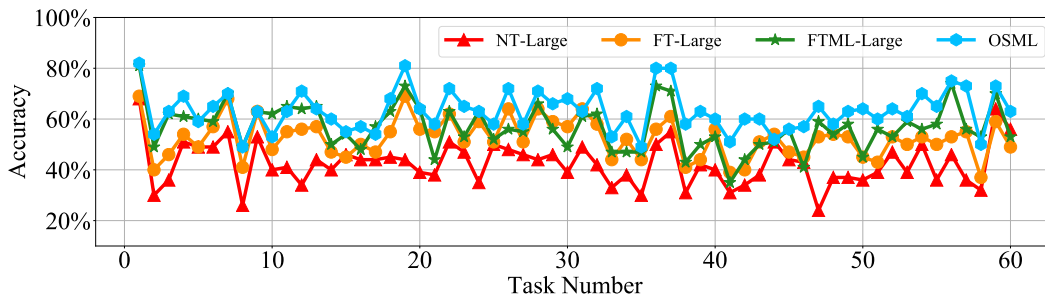

| Models | Blur Acc. | Night Acc. | Pencil Acc. | Overall Acc. | AR |
|---|---|---|---|---|---|
| NT | $49.80 \pm 3.91\%$ | $47.70 \pm 2.91\%$ | $47.55 \pm 5.18\%$ | $53.32 \pm 2.30\%$ | - |
| NT-Large | $43.05 \pm 3.99\%$ | $41.30 \pm 2.66\%$ | $43.25 \pm 4.24\%$ | $42.53 \pm 2.15\%$ | 3.92 |
| FT | $51.50 \pm 4.90\%$ | $49.00 \pm 3.82\%$ | $50.90 \pm 5.30\%$ | $50.47 \pm 2.73\%$ | - |
| FT-Large | $54.60 \pm 2.99\%$ | $50.35 \pm 2.64\%$ | $52.45 \pm 3.92\%$ | $52.46 \pm 1.91\%$ | 2.82 |
| FTML | $58.90 \pm 3.52\%$ | $56.40 \pm 3.53\%$ | $54.60 \pm 5.46\%$ | $56.63 \pm 2.50\%$ | - |
| FTML-Large | $57.55 \pm 3.76\%$ | $56.70 \pm 3.92\%$ | $56.30 \pm 4.05\%$ | $56.85 \pm 2.26\%$ | 2.13 |
| **OSML** | $\mathbf{64.10 \pm 3.12\%}$ | $\mathbf{65.25 \pm 3.24\%}$ | $\mathbf{60.35 \pm 3.65\%}$ | $\mathbf{63.23 \pm 2.00\%}$ | **1.13** |

Figure 6: Comparison between OSML with baselines with increased model capacity. We list orignial NT, FT, FTML are listed for comparison without providing AR.

**Learning Efficiency Analysis.** In heterogeneous datasets, the performances fluctuate across different tasks due to the non-overlapped classes. Similar to [10], the learning efficiency is evaluated by the number of samples in each task. We conduct the learning efficiency analysis by varying the training samples and report the performance in Table 1. Here, two baselines (FTML and DPM) are selected for comparison. In this table, we observe that OSML is able to consistently improve the performance under different settings. The potential reason is that selecting meaningful meta-knowledge pathway captures the heterogeneous task information and further improve the learning efficiency. For the homogeneous data, we analyze the amount of data needed to learn each task in Appendix D and the results indicate that the ability of OSML to efficiently learn new tasks.

Table 1: Performance w.r.t. the number of samples per task on Meta-dataset.

| # of Samples | Models | Aircraft Acc. | Flower Acc. | Fungi Acc. | Overall Acc. |
|---|---|---|---|---|---|
| 200 | FTML | $55.92 \pm 3.13\%$ | $54.65 \pm 4.52\%$ | $45.69 \pm 2.97\%$ | $52.08 \pm 2.51\%$ |
| | DPM | $57.15 \pm 3.29\%$ | $53.34 \pm 5.38\%$ | $44.33 \pm 2.77\%$ | $51.60 \pm 2.79\%$ |
| | **OSML** | $\mathbf{60.18 \pm 2.89\%}$ | $\mathbf{58.28 \pm 4.80\%}$ | $\mathbf{48.25 \pm 3.02\%}$ | $\mathbf{55.57 \pm 2.59\%}$ |
| 300 | FTML | $62.88 \pm 3.10\%$ | $58.37 \pm 5.02\%$ | $47.96 \pm 2.49\%$ | $56.40 \pm 2.59\%$ |
| | DPM | $64.43 \pm 3.26\%$ | $59.72 \pm 5.37\%$ | $48.10 \pm 2.60\%$ | $57.42 \pm 2.83\%$ |
| | **OSML** | $\mathbf{66.57 \pm 3.27\%}$ | $\mathbf{65.07 \pm 4.38\%}$ | $\mathbf{53.72 \pm 2.81\%}$ | $\mathbf{61.78 \pm 2.63\%}$ |
| 400 | FTML | $65.85 \pm 3.85\%$ | $59.05 \pm 4.83\%$ | $48.00 \pm 2.78\%$ | $57.63 \pm 2.93\%$ |
| | DPM | $66.67 \pm 4.07\%$ | $63.40 \pm 4.89\%$ | $50.15 \pm 3.53\%$ | $60.07 \pm 3.02\%$ |
| | **OSML** | $\mathbf{67.99 \pm 3.52\%}$ | $\mathbf{68.55 \pm 4.59\%}$ | $\mathbf{58.45 \pm 2.89\%}$ | $\mathbf{65.00 \pm 2.46\%}$ |

## 5   Discussion with Related Work

In meta-learning, the ultimate goal is to enhance the learning ability by utilizing and transferring learned knowledge from related tasks. In the traditional meta-learning setting, all tasks are generated from a stationary distribution. Under this setting, there are two representative lines of meta-learning algorithm, including optimization-based meta-learning [7–9, 13, 11, 17, 19, 26, 28, 31] and non-parametric meta-learning [12, 22, 24, 34, 35, 38, 43]. In this work, we focus on the optimization-based meta-learning. Recently, a few studies consider non-stationary distribution during the meta-testing phase [2, 25], while the learned meta-learner is still fixed after the meta-training phase. Finn *et al.* [10] further handles the non-stationary distribution by continuously updating the learned prior. Unlike this study that using the shared meta-learner, we investigate tasks are sampled from heterogeneous distribution, where the meta-learner are expected to be capable of non-uniform transferring.

To handle the heterogeneous task distribution, under stationary setting, a few studies modulate the meta-learner to different tasks [3, 27, 39, 41, 42]. However, the performance of modulating mechanism depends on the reliability of task representation network, which requires a number of training tasks and is impractical in online meta-learning setting. Jerfel *et al.* [15] further bridge optimization-based meta-learning and hierarchical Bayesian and propose Dirichlet process mixture of hierarchical Bayesian model to capture non-stationary heterogeneous task distribution. Unlike this work, we encourage layer-wise knowledge block exploitation and exploration rather than create a totally new meta-learner for the incoming task with dissimilar information, which increases the flexibility of knowledge sharing and transferring.

The online meta-learning setting is further related to the continual learning setting. In continual learning, various studies focus on addressing catastrophic forgetting by regularizing the parameter changing [1, 4, 16, 32, 44], by expanding the network structure [6, 18, 20, 30], by maintaining a episodic memory [5, 23, 29, 33]. Li *et al.* [20] has also considered settings that expanding the network in a block-wise manner, but have not focused on forward transfer and not explicitly utilized the task-specific adaptation. In addition, all continual learning studies limit on a few or dozens task, where the online meta-learning algorithms enable agents to learn and transfer knowledge sequentially from several tens or hundreds of related tasks.

## 6   Conclusion and Discussion

In this paper, we propose OSML – a novel framework to address online meta-learning under heterogeneous task distribution. Inspired by the knowledge organization in human brain, OSML maintains a meta-hierarchical structure that consists of various knowledge blocks. For each task, it constructs a meta-knowledge pathway by automatically select the most relevant knowledge blocks. The information from the new task is further incorporated into the meta-hierarchy by meta-updating the selected knowledge blocks. The comprehensive experiments demonstrate the effectiveness and interpretability of the proposed OSML in both homogeneous and heterogeneous datasets. In the future, we plan to investigate this problem from two aspects: (1) effectively and efficiently structuring the memory buffer and storing the most representative samples for each task; (2) theoretically analyzing the generalization ability of proposed OSML; (3) investigating the performance of OSML on more real-world applications.

## Broader Impact

The rapid development of information technology has greatly increased the machine's ability to continuously and quickly adapt to the new environment. For example, in an autonomous driving scenario, we need to continuously allow the machine to adapt to the new environment. Without such ability, autonomous cars are difficult to be applied to real scenarios, and may also cause potential safety hazards. In this paper, we mainly study the continuous adaptation of meta-learning to complex heterogeneous tasks. Compared to homogeneous tasks, heterogeneous tasks are not only more common in the real world, but also more challenging.

Investigating this problem benefits the improvement of learning ability under the online meta-learning setting, which further greatly benefits a large number of applications. Especially, the meta-hierarchical tree we designed can capture the structural association between different tasks. For example, in the disease risk prediction problem, we expect that the agent is capable of continuously adjusting the model to adapt to different diseases. Considering the great correlation between diseases, our model is able to automatically detect these correlations and incorporate the rich external knowledge (e.g., medical knowledge graph).

## Acknowledgement

The work was supported in part by NSF awards #1652525 and #1618448. The views and conclusions contained in this paper are those of the authors and should not be interpreted as representing any funding agencies.

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
