[Supplementary Material]

# Online Structured Meta-learning – Appendix

**Huaxiu Yao**[1]*, **Yingbo Zhou**[2], **Mehrdad Mahdavi**[1]
**Zhenhui Li**[1], **Richard Socher**[2], **Caiming Xiong**[2]
[1]Pennsylvania State University, [2]Salesforce Research
[1]{huaxiuyao,mzm616,zul17}@psu.edu, [2]{yingbo.zhou,cxiong}@salesforce.com, richard@socher.org

## A  Experimental Settings

### A.1  Homogeneous Dataset

Follow the traditional meta-learning setting [1, 4, 5], each knowledge block is comprised of a convolution layer, a batch normalization layer, and a ReLU activation. We set the learning rate $\beta_1$ - $\beta_5$ as 0.001, 0.01, 0.001, 0.01, 0.001, respectively. For each class, the number of both support and query samples are set as 40 and the number of test samples is 20. Note that, the final linear layer is randomly re-initialized for each task.

### A.2  Heterogeneous Datasets

Follow [3], Multi-filtered miniImagenet is constructed by separately applying three widely-used filters (i.e.,Blur, Night and Pencil) on traditional miniImagenet. Figure 1 gives the effect of different filter on the same image. For both Multi-filtered miniImagenet and Meta-dataset, the structure of each knowledge block is the same as Rainbow MNIST. The last linear layer is also randomly re-initialized for each task. The learning rates $\beta_1$ - $\beta_5$ for both Multi-filtered miniImagenet and Meta-dataset are set as 0.001, 0.01, 0.001, 0.01, 0.001. respectively. For each class, both support size and query size are set as 40. The number of leaved testing data samples is set as 20 for each class.

(a) : Original   (b) : Blur   (c) : Night   (d) : Pencil

Figure 1: Images with different filters.

## B  Analysis of Constructed Meta-pathway on Multi-filtered miniImagenet

In this section, we show the constructed meta-pathway on multi-filtered miniImagenet by illustrating the selected knowledge-blocks in Figure 2. Similar to the observations on Meta-dataset, some knowledge blocks in Multi-filtered miniImagenet are dominated by a specific sub-dataset (e.g., knowledge block 4 in layer 2). Additionally, we observe that Blur and Night are more likely to share knowledge blocks than Blur-Pencil or Night-Pencil. The potential reason is that Blue and Night maintain more information from the original images than Pencil, which makes them more similar.

(a) : Layer 1      (b) : Layer 2      (c) : Layer 3      (d) : Layer 4

Figure 2: Frequency of selected knowledge blocks for each sub-dataset on Multi-filtered miniImagenet. Figure (a)-(d) illustrate the frequency in layer 1-4.

## C    Analysis about Model Capacity on Meta-dataset

In this section, we conduct the experiments on Meta-dataset and show the summarized performance with learning curve in Figure 3. In this Figure, first, we observe that both NT-Large and FT-Large outperforms NT and FT accordingly. The potential reason is that pre-trained model is used in Meta-dataset and thereby increasing the model capacity enhance the representation ability rather than cause overfitting. Second, compared OSML with other baselines, similar to the performance on Multi-filtered miniImagenet, the consistent improvements further demonstrate that the improvements of OSML stems from the exploitation of structured information rather than larger representation capacity.

| Models | Aircraft Acc. | Flower Acc. | Fungi Acc. | Overall Acc. | AR |
|---|---|---|---|---|---|
| NT | $59.40 \pm 3.97\%$ | $60.15 \pm 5.23\%$ | $46.15 \pm 3.57\%$ | $55.23 \pm 2.98\%$ | - |
| NT-Large | $64.75 \pm 3.80\%$ | $64.50 \pm 4.64\%$ | $52.45 \pm 3.48\%$ | $60.56 \pm 2.73\%$ | 2.86 |
| FT | $66.45 \pm 3.80\%$ | $52.20 \pm 4.55\%$ | $42.90 \pm 3.45\%$ | $53.85 \pm 3.35\%$ | - |
| FT-Large | $65.95 \pm 3.91\%$ | $60.55 \pm 4.58\%$ | $52.40 \pm 3.18\%$ | $59.63 \pm 2.67\%$ | 2.76 |
| FTML | $65.85 \pm 3.85\%$ | $59.05 \pm 4.83\%$ | $48.00 \pm 2.78\%$ | $57.63 \pm 2.93\%$ | - |
| FTML-Large | $\mathbf{68.20 \pm 4.21\%}$ | $59.75 \pm 4.98\%$ | $51.75 \pm 3.27\%$ | $59.90 \pm 2.96\%$ | 2.72 |
| **OSML** | $\mathbf{67.99 \pm 3.52\%}$ | $\mathbf{68.55 \pm 4.59\%}$ | $\mathbf{58.45 \pm 2.89\%}$ | $\mathbf{65.00 \pm 2.46\%}$ | **1.65** |

Figure 3: Top: performance of all tasks generated from Meta-dataset. Bottom: Comparison between OSML with NT, FT, FTML with increased model capacity.

## D    Task Efficiency of Homogeneous Dataset (Rainbow MNIST)

In this section, we analyze task learning efficiency on the homogeneous dataset (i.e., Rainbow MNIST). Specifically, we follow [2] to analyze the amount of data needed to learn each task and show the results in Figure 4. In our experiment, since some tasks are not able to reach the target accuracy (slightly below the target accuracy) even using all data samples, we calculate the number of samples as the whole dataset for these tasks. We observe that our method requires less number of samples to reach the target accuracy as compared to the other methods in most cases, which indicates our ability to efficiently learn new tasks.

Figure 4: Learning efficiency analysis on Rainbow MNIST data.

## Footnotes

*Part of the work was done while the author interned at Salesforce Research