[Reviews · NeurIPS 2020]

Review 1

Summary and Contributions: The authors propose an online structured meta-learning (OSML) framework. They unravel the meta-learner as a meta-hierarchical graph with multiple structured knowledge blocks, where each block represents one type of structured knowledge. When a new task arrives, it searches for the most relevant knowledge blocks and constructs a meta-knowledge pathway in this meta-hierarchical graph. Besides the meta-knowledge pathway, it also organizes the newly learned experience.

Strengths: + Heterogeneous and non stationary distribution introduction towards online meta-learning. + Knowledge graph of the meta-learner, selecting knowledge pathway for on task., and add new blocks corresponding to task information. + Experiments on both homogeneous and heterogeneous distribution.

Weaknesses: - The proposed seems to be combination of [20] and [9]. - results on Aircraft and Fungi are not convincing, because of higher 95% confidence interval they fall into the same range. - I think other evaluation metrics such as task learning efficiency would be a better comparison which are reported in [9] should be considered given my first concern about the results. - How does OSML perform with lesser data points? - An ablation study with varying number of datapoints would make the paper more convincing

Correctness: Empirically, the method performs better than the reported baselines. However, as mentioned above, couple of metrics and new studies could make the paper more convincing. To search for the pathway, they follow DARTS[1], and to update knowledge block and add new information, the follow FTML[2] to iteratively update the knowledge block parameters. [1] DARTS: DIFFERENTIABLE ARCHITECTURE SEARCH, ICLR 2019 [2] Online Meta-Learning, ICML 2019

Clarity: The paper is easy to read.

Relation to Prior Work: The authors have mentioned the prior work on online meta-learning and mentioned that most of that work consider stationary distribution in this setting. They propose a different setting where the tasks come from non-stationary distribution.

Reproducibility: Yes

Additional Feedback: Please, correct the Typo in figure 5 knoledge -> knowledge.


Review 2

Summary and Contributions: This paper presents an approach to online meta-learning known as Online Structured Meta-learning (OSML). It uses a collection of knowledge blocks with a gating mechanism to select which "knowledge" to use for a given task. The authors show good performance on modifications of standard meta-learning benchmark tasks. The contribution of this paper lies in the new architecture, and the improved performance as a result of this architecture.

Strengths: The central strength of this paper lies in its performance; the authors clearly show that the method quantitatively outperforms reasonable benchmarks in the online meta-learning setting. The authors also address the difference in number of parameters and show that the baselines, adjusted to a have a comparable number of parameters, are outperformed by OSML.

Weaknesses: The primary weaknesses of the approach is that the authors do not sufficiently convince the reader that the knowledge block approach is necessary to yield the achieved performance improvements. Generally, it is unclear if the machinery of the knowledge blocks truly result in modular knowledge representations. While the authors present a framework that achieves reasonably strong performance on their chosen benchmarks, I think the paper does not effectively present a contribution that authors could build on in future work. I believe ablations of the model would help convince a reader of the utility of the knowledge block formulation. In particular, an experiment with the same architecture but without the meta-knowledge pathway construction would help convince a reader that this step is necessary. The homogeneous vs heterogeneous experiments are quite unclear. In [14] the authors sequentially experience new tasks (corresponding to new filters) in a more typical continual learning setting. In particular, in that setting, the model has extensive experience in one task before moving to another. As far as I can tell in this work, the different artistic filters are sampled for different tasks, and so these are mixed throughout training. As such, it is unclear what the contribution of the heterogeneous experiments is over the homogeneous tasks.

Correctness: The paper appears to be correct.

Clarity: The paper is reasonably clear, with the exception of the motivation of some of the experiments as discussed above.

Relation to Prior Work: The approach taken in this paper is fundamentally a modular approach to meta-learning. The authors should reference https://arxiv.org/abs/1806.10166 as well as subsequent work, with which OSML has strong similarities. Other than this, the paper appears to position itself fairly clearly with respect to the existing literature.

Reproducibility: Yes

Additional Feedback: I would be willing to revise my score upward if clear experiments motivating the knowledge block formulation (as discussed above) are presented. ------ After Rebuttal: Thank you to the authors for your response. I am still concerned that the evaluation of the meta-pathway construction is not sufficiently thorough. In particular, I believe an ablation of the model without the argmax step would be a good comparison (maintain the "soft" weighted sum of blocks). Moreover, as the other reviewers have noted, this method is relatively straight-forward combination of existing ideas in the community. I have increased my score, but still believe more thorough evaluation of the claims is necessary.


Review 3

Summary and Contributions: This paper proposed a novel method for online meta-learning that relies on the notion of meta-learning knowledge that can be combined when encountering a new task. While the problem of task heterogeneity has been tackled before in the literature, to the best of my knowledge this is a first attempt to do so in an online setting. --- Post rebuttal update --- The authors provide a strong rebuttal that addressed my main concerns. In particular, that a non-trivial amount of pathways are indeed being constructed and that this matters. New experimental results comparing hierarchical clustering provide further evidence that the proposed mechanism for modularisation is promising avenue for further research. As such, I've increased my score.

Strengths: The paper follows [1] closely and differs in architectural design. Their proposed method, OSML, introduces an architecture where each layer is made up of a set of blocks. The number of blocks can grow dynamically with time. For each task, a search mechanism goes through the architecture layer by layer and selects what block in what layer to use, or alternatively selects a new randomly initialised block. Once the search process is complete, the selected blocks in each layer are used to construct a task-specific learner. This learner is then fine-tuned to the task. After each task, a MAML-style objective taken from [1] is used to update the initial representations in the blocks chosen for the given task. I find the idea interesting and new as far as I know, with encouraging empirical results. Hence I recommend this paper for publication. With that said, I believe the paper can do a better job placing its contributions in light of prior works. [1] Finn et. al., Online Meta-Learning. 2019.

Weaknesses: I have some qualms with the way a pathway is being selected. The proposed method uses a softmax distribution to obtain importance weights for each block in each layer. However, rather than sampling from this distribution, which would allow the model a means of exploration, the authors use the argmax. Hence the model is always greedy, and because the softmax distribution is constructed by evaluating each block 0-shot on a task, a randomly initialised block is exceedingly unlikely to selected. Hence the proposed architecture seems bottlenecked in its ability to actually spin up new blocks on the fly. There are no ablation studies in this paper, which I believe could help enhance the paper. In particular, it would be valuable to understand trade-offs involved with the constructing a new block or using existing but tuned ones.

Correctness: Claims and methodology is largely correct, except in one minor case. In practice, the authors side-step the MAML objective and use a “first-order approximation”. However the approximation the authors use is not of MAML, instead the authors are implicitly using Reptile [2] which significantly changes the interpretation of the meta-updates (see [3]). [2] Nichol et. al., On First-Order Meta-Learning Algorithms. 2018. [3] Flennerhag et. al., Transferring Knowledge Across Learning Processes. 2018.

Clarity: The paper is largely well written and easy to follow.

Relation to Prior Work: Meta-learning under task heterogeneity is not new, and this paper does not only superficially connect to that literature. While relevant citations are there, they are mentioned in sweeping sentences that clobber all citations into one long list, without any detail as to how they relate to the proposed method whatsoever. Similarly, the idea of picking paths through a network has a rich history that is only mentioned in passing. As it currently stands, this paper gives the impression of greater novelty that can be claimed - a more transparent comparison to previous work would help contextualize the contributions of this paper.

Reproducibility: Yes

Additional Feedback:


Review 4

Summary and Contributions: To sequentially learn tasks from a non-stationary and potentially heterogeneous task distribution, this paper proposed an Online Structured Meta-learning model (OSML) that explicitly disentangles the meta-learner as a meta-hierarchical graph with different knowledge blocks and do adaptation via both meta-knowledge pathway selection and new block exploration. Experimental results showed that the proposed model can better handle the non-stationary and potentially heterogeneous task distribution scenarios via the dynamic changes of the meta-learner than several listed baselines.

Strengths: 1. This paper is well organized and the proposed method is clearly demonstrated. 2. The problem this paper focused on is worth studying and the authors proposed an elegant way to enable dynamic changes of the meta-learner to perform better in a scenario where tasks are from non-stationary and heterogeneous distribution. 3. The proposed model surpasses several baselines by a large margin, which strongly proves the effectiveness of the proposed OSML model.

Weaknesses: 1. The novelty is somewhat limited. The motivation and the model structure share similar spirits with Hierarchically Structured Meta-learning (HSML), which is published on ICML’19. More thorough discussions are needed to demonstrate how the contribution of this paper differs from HSML. 2. The baselines are not complete. The proposed method is related to both online meta-learning and structured meta-learning, but the authors only provide online meta-learning baselines, ignoring structured meta-learning models (i.e., Hierarchically Structured Meta-learning). 3. Though this paper provided some analyses about the constructed meta-hierarchical tree, more detailed illustrations are recommended for better understanding and explanation.

Correctness: Yes, the proposed method sounds reasonable.

Clarity: Yes.

Relation to Prior Work: Although the authors did discuss the differences between this work and the research line of online meta-learning and involve them as baselines, they did not clearly demonstrate the differences between this work and several structured meta-learning models (i.e., Hierarchically Structured Meta-learning, published on ICML’19), nor did they provide any baselines of this group. In fact, HSML not only proposed a hierarchical task clustering structure which also tackles task uncertainty and heterogeneity issues but also proved its capability of continual adaptation. More discussions about the differences as well as additional baselines of structured meta-learning are suggested.

Reproducibility: Yes

Additional Feedback:

[Author Response · NeurIPS 2020]

We sincerely thank all reviewers for their feedback. All experiments are conducted on Meta-dataset and averaged
performance of each group is reported. Ai, Fl, Fu denote Aircraft, Flower, and Fungi, respectively. The performances of
FTML and DPM are also reported for comparison. We will correct typos, add the suggested references with discussions,
and improve the overall presentation.

**To Reviewer 1**
**- Regarding novelty:** We propose a generic framework for doing online meta-learning that not only works under
homogeneous but also heterogeneous settings. We evaluated our work under the same setting as compared to [9]. We
choose to use [20] as an implementation for knowledge pathway construction. Our method is flexible and one could
choose to use other method in this step, e.g. evolutionary or reinforcement learning strategy.
**- Results of fungi** (58.45 $\pm$ 2.89%) is significant better than best baseline (DPM: 50.15 $\pm$ 3.53%).
The comments on Aircraft result is true, and we have discussed it in L210 - 214 in the submission.
**- Task learning efficiency and Performance v.s. # of samples:** In heterogeneous
datasets, the performances fluctuate across different tasks due to non-overlapped classes.
Similar to [20] (Fig 4), learning efficiency is evaluated by # of samples. The performance
of OSML w.r.t. # of samples are reported in Table 1, showing that OSML is able to
consistently improve performance. For Rainbow MNIST, we follow [9] to analyze the
amount of data needed to learn each task and show the results in Figure 1. Notice that
our method requires less number of samples to reach the target accuracy as compared
to the other methods, which indicates our ability to efficiently learn new tasks.

| Model-# Data | Ai | Fl | Fu |
|---|---|---|---|
| FTML-20 | 55.92 | 54.65 | 45.69 |
| DPM-20 | 57.15 | 53.34 | 44.33 |
| OSML-20 | **60.18** | **58.28** | **48.25** |
| FTML-30 | 62.88 | 58.37 | 47.96 |
| DPM-30 | 64.43 | 59.72 | 48.10 |
| OSML-30 | **66.57** | **65.07** | **53.72** |
| FTML-40 | 65.85 | 59.05 | 48.00 |
| DPM-40 | 66.67 | 63.40 | 50.15 |
| OSML-40 | **67.99** | **68.55** | **58.45** |

Figure & Table 1

**To Reviewer 2**
**- Importance of pathway construction:** We have conducted two experiments to show
the effectiveness of meta-knowledge pathway selection. First, fine-tuning (FT) is used
in Fig. 2 – 4, where only one knowledge block each layer is involved and fine-tuned for
each task. This corresponds to the case where no meta-knowledge pathway is getting
constructed. Second, we also use FTML with the same number of parameters as the final
number of blocks in the meta-hierarchical graph (i.e., same architecture) and discuss
the performance in L227-238. OSML consistently outperforms these two baselines.
**- Homogeneous v.s. Heterogeneous Settings:** We follow the setting that is suggested
in [38,40,41]: (1) non overlap between classes; (2) the underlying distribution is mul-
timodal. The suggested setting from [14] is also valuable and we conduct additional
experiment to evaluate it by feeding Fl->Ai->Fu. The performance obtained are [FTML: (Ai66.51 | Fl67.28 | Fu50.89),
DPM: (Ai68.85 | Fl68.90 | Fu53.20), OSML: (Ai71.35 | Fl72.09 | Fu57.00)], demonstrating the effectiveness of OSML.
**To Reviewer 3**
**- New block bottleneck:** When a new task arrives, we randomly select two sets $\mathcal{D}_t^{supp}$ and $\mathcal{D}_t^{query}$ to search the
functional regions (L 4-5 in Alg. 1). For the new initialized blocks, motivated by [20], it can access the corresponding
data and be tuned at this time. Therefore, the new block is possible to get selected.
**- Ablation study:** We have designed two models to show the importance of new block construction and path selection.
Please kindly refer to "Importance of pathway construction" of Reviewer 2.
**- Clarification of first-order approximation:** We use FOMAML [7] rather than Reptile for first-order approximation,
where the loss of FOMAML is calculated on query sets with adapted parameters.
**- Discussion with previous work:** We have discussed five recent studies that focus on task heterogeneity in both
Introduction (L36-41) and Related Work (L250-258). The list of citations (i.e., [26,38,40,41]) focus on the stationary
scenario. [14] focuses on non-stationary task distribution and has been discussed in detail.
**To Reviewer 4**
**- Discussion with HSML:** The major difference between OSML and HSML are two-fold: (1) The settings are different.
HSML focuses on the stationary task distribution while OSML tackles the challenge of non-stationary setting. The
continual adaptation setting is evaluated under the stationary scenario. In Figure 4 and 7, curves show meta-training
performance and tables report stationary meta-testing results. (2) HSML captures task heterogeneity by customizing the
model initialization using task-wise cluster representation. While OSML encourages the layer-wise knowledge block
exploitation and exploration, which improves the flexibility of knowledge transfer.
Though the original HSML is evaluated on stationary scenario, to make comparison we have evaluated the model under
our setting by introducing task-aware parameter customization and hierarchical clustering structure. The performances
are (Ai64.33 | Fl62.75 | Fu52.18) compared to (Ai67.99 | Fl68.55 | Fu58.45) shown in table 4 in our paper, which further
verifies the effectiveness of OSML.

[Meta-Review · NeurIPS 2020]

This paper convinced the reviewers that the modular approach to meta-learning proposed therein outperforms rivals in a series of experiments. There were concerns about novelty, as the approach can be seen as a combination of prior approaches, but honestly I don't see why this should be seen as a negative. It is not the duty of scientific progress to surprise us with regard to the sources from where it is drawn. If combining existing approaches exploits synergies and complementarities, then a paper demonstrating the empirical success of such a combination is worth sharing with the community. Accept.